# Improved Image Augmentation for Convolutional Neural Networks by Copyout and Copy-Pairing

## Abstract

Image augmentation is a widely used technique to improve the performance of convolutional neural networks (CNNs). In common image shifting, cropping, flipping, shearing and rotating are used for augmentation. But there are more advanced techniques like *Cutout* and *SamplePairing*.

In this work we present two improvements of the state-of-the-art Cutout and SamplePairing techniques. Our new method called *Copyout* takes a square patch of another random training image and copies it onto a random location of each image used for training. The second technique we discovered is called *CopyPairing*. It combines Copyout and SamplePairing for further augmentation and even better performance.

We apply different experiments with these augmentation techniques on the CIFAR-10 dataset to evaluate and compare them under different configurations. In our experiments we show that *Copyout* reduces the test error rate by 8.18% compared with Cutout and 4.27% compared with SamplePairing. CopyPairing reduces the test error rate by 11.97% compared with Cutout and 8.21% compared with SamplePairing.

Copyout and CopyPairing implementations are available at https://github.com/anonym/anonym.

## 1 Introduction

Image augmentation is a data augmentation method that generates more training data from the existing training samples. This way the neural network is trained with more different images. This helps to generalize better on new and unknown images and leads to better performance with smaller datasets. Image augmentation is especially useful in domains where training data is limited or expensive to obtain like in biomedical applications.

Image augmentation is a regularization technique like Dropout (Hinton et al., 2012) and weight regularization (Ng, 2004). But it is different in one way. It is not part of the network itself which makes it relatively easy and cheap to apply. Images can be augmented on runtime. While the GPU runs the neural network, the CPU can do the augmentation for the next batch in an asynchronous manner.

Two state-of-the-art techniques for advanced image augmentation are called Cutout (Devries & Taylor, 2017) and SamplePairing (Inoue, 2018). In our work we show two improvements to these techniques. Our main contributions are:

1. Copyout image augmentation described in section 2

2. CopyPairing image augmentation described in section 3

3. Several experiments to compare them in Section 4.3 ff.

4. An analysis of the reasons why these methods improve performance in Section 5

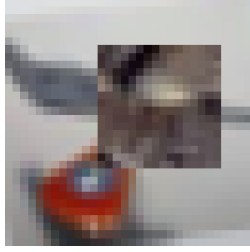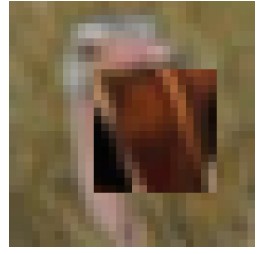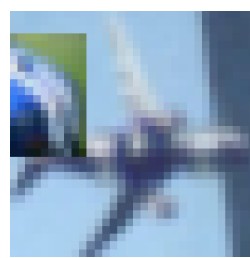

Figure 1: Examples of CIFAR-10 (Krizhevsky, 2009) Images augmented by Copyout.

## 2 COPYOUT

The Copyout image augmentation technique should be applied on runtime on each image before it is used to build the mini-batch for training. The results must not be stored and reused. This would reduce the variance of the training images and reduce the performance of the CNN. The augmentation is conducted in these three steps:

1. Select a target training image to augment. On this target image select a square area on a random location. By this way it is possible that the randomly selected square leaves the image on one or two sides. This can effectively change the shape of the square to a rectangle. As stated by Devries & Taylor (2017) this is an important feature and improves performance. In our implementation the center of the square never leaves the target image.

2. Randomly select a source image from the training set. On this source image also select an area of the same extend as the one we selected on the target image. This second area is also selected on a random location but must be fully placed within the source image.

3. Copy the area of the source image to the target image.

The extend of the square is a hyperparameter which must be set before training starts and does not change during the training process. Apparently the extend must be smaller than the training images. Devries & Taylor (2017) found that for Cutout the extend of the square is more important than the shape. This is the reason why we also decided to use squares for simplicity.

Examples of images augmented by Copyout are given in Figure 1.

## 3 COPYPAIRING

CopyPairing is a mixture of Copyout and SamplePairing (Inoue (2018) and Appendix A.2). CopyPairing follows the same three phases as SamplePairing. The only difference is that it replaces the phases where SamplePairing is disabled by using Copyout. CopyPairing is conducted by the following three steps:

1. Only Copyout is enabled and SamplePairing is completely disabled for the first epochs.

2. SamplePairing is enabled for a few epochs and then Copyout is used for the next few epochs. This is alternating for the majority of the epochs.

3. At the end of training only Copyout is enabled while SamplePairing is completely disabled again. This is also called fine-tuning.

## 4 EXPERIMENTS

### 4.1 EXPERIMENTAL SETUP

The CNN we use for our experiments consists of six 2D convolution layers with Glorot uniform kernel initializer (Glorot & Bengio, 2010), Rectified Linear Unit (ReLU) activation (Glorot et al., 2011) and Batch Normalization (Ioffe & Szegedy, 2015). Two dense layers are applied at the output

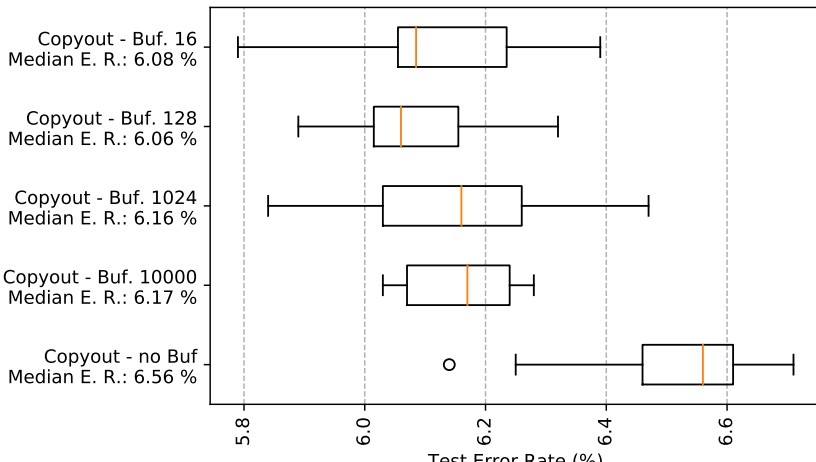

Figure 2: Comparison of Copyout with different buffer sizes to store augmented images. "Copyout - no Buf" selects the source images from the whole training set without buffer and augmentation. Method name with buffer size and median test error rate is given by the caption. Each experiment has been repeated at least 12 times to draw the boxplots.

and combined with two Dropout layers (Hinton et al., 2012). Although Ioffe & Szegedy (2015) propose to place the Batch Normalization layer in front of the activation function we found out that this CNN performs better when activation and Batch Normalization are interchanged. Appendix A.1 Listing 1 presents the definition of the concrete CNN with Keras (Chollet et al., 2015).

As the optimizer we use Adam with parameters as provided in the paper Kingma & Ba (2014) (learning rate of 0.001 e.g.). The mini-batch size is 128 and we train for 1200 epochs. The CIFAR-10 Dataset (Krizhevsky, 2009) is used for reference and we try to minimize the test error rate in our experiments.

## 4.2 BASIC AUGMENTATION

For basic augmentation we randomly rotate the images by up to 15 deg. to the left and right, randomly flip them horizontally and randomly zoom them by a factor of up to 0.2. Appendix A.1 Listing 2 presents our implementation of basic augmentation with the Keras `ImageDataGenerator` class.

It is important where and how basic augmentation is applied. When it is only applied to the target image and not the source image the performance will be reduced. It is important that the source image is also augmented. This can be achieved when basic augmentation is applied after the square has been copied. But we selected another method.

Our basic augmentation is combined with Copyout and SamplePairing by using a buffer. We apply basic augmentation on the target image and store it in a buffer. For Copyout and SamplePairing we then select the source image from this buffer randomly. This way we have augmentation for source and target images.

Figure 2 displays different experiments with varying buffer sizes. It shows that the buffer size can be small without reducing performance. It also shows that augmentation of source images is important. The "Copyout - no Buf" experiment selects the source images from the whole training set without taking them from the buffer and without applying basic augmentation on the source images. Its median test error rate is significantly increased.

## 4.3 METHOD COMPARISON

We compared the different augmentation techniques while using the setup described in Section 4.1. All experiments have been combined with basic augmentation (see Section 1). Each experiment was

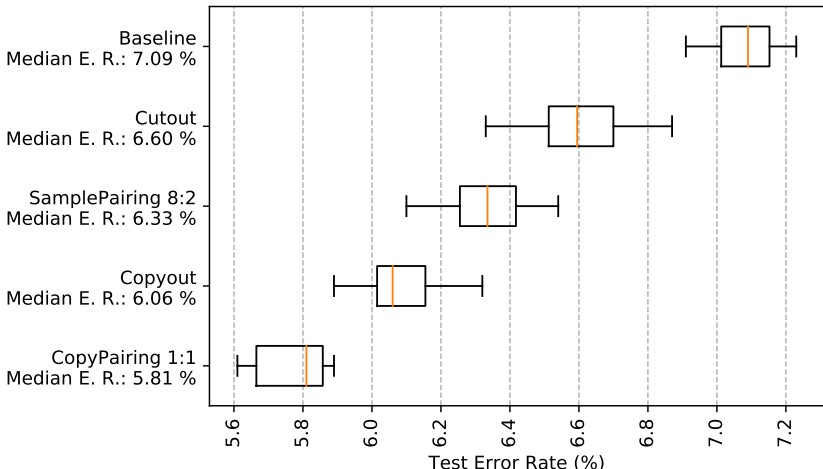

Figure 3: Comparison of Cutout, SamplePairing, Copyout and CopyPairing. Also illustrated is the baseline with just default image augmentation and without any further augmentation. Method name and median test error rate is given by the caption. Each experiment has been repeated at least 12 times to draw the boxplots.

repeated at least 12 times to get meaningful boxplots. Boxplots with test error rates are shown in Figure 3.

For the *baseline* experiment we only used basic augmentation and reached a median test error rate of 7.09%. With *Cutout* augmentation we reached a median test error rate of 6.6%.

For the *SamplePairing* experiment we used the following configuration: In phase 1 we disabled SamplePairing for the first 100 epochs. Phase 2 lasted 900 epochs, SamplePairing was turned on for 8 epochs and turned off for 2 epochs alternating. Phase 3 (fine-tuning) lasted the remaining 200 epochs. This corresponds to the setup of Inoue (2018).
With SamplePairing we reached a median test error rate of 6.33%.

For *Copyout* augmentation we selected a 16 pixel extend of the square as hyperparameter and reached a median test error rate of 6.06%. This is an 8.18% improvement compared with Cutout and a 4.27% improvement compared with SamplePairing.

The experiment with *CopyPairing* was done with the following configuration: For the Copyout part we selected a 16 pixel extend of the square as hyperparameter. In phase 1 we disabled the SamplePairing part for the first 100 epochs with only Copyout enabled. Phase 2 lasted 900 epochs with alternating SamplePairing and Copyout part for one epoch each. This was executed alternating. Phase 3 (fine-tuning) lasted the remaining 200 epochs with the SamplePairing part turned off and only Copyout enabled.
We reached a median test error rate of 5.81%. This is an 11.97% improvement compared with Cutout and an 8.21% improvement compared with SamplePairing. Even compared with the Copyout augmentation we reached an improvement of 4.13%.

### 4.4 SQUARE AREA EXTENT COMPARISON FOR COPYOUT

In our experiments with Copyout we selected a 16 pixel extend of the square as hyperparameter. To examine if the selection was the best choice we executed three different experiments. One with 16 pixel extend, one with 15 and 17 pixel extends. The result is that larger and smaller extend numbers for the square are lowering the performance for our experimental setting and that a 16 pixel extend is the best choice for this setup (see Figure 4).

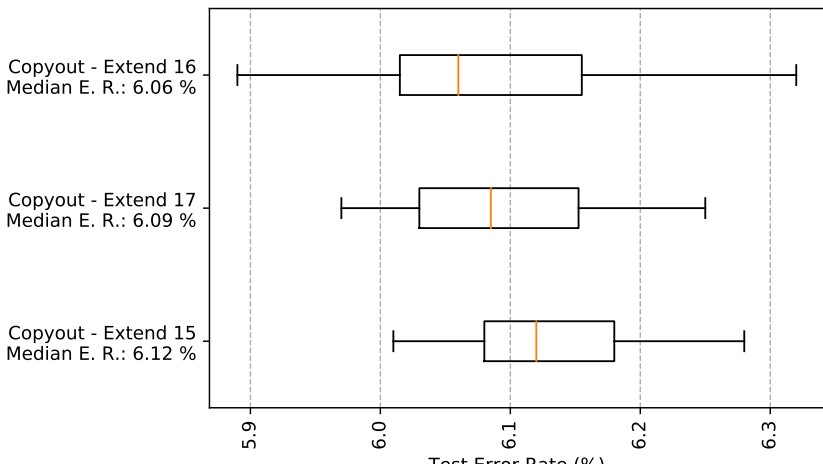

Figure 4: Comparison of three different extends of the square area for Copyout. Method name with the selected extend and median test error rate is given by the caption. Each experiment has been repeated at least 12 times to draw the boxplots.

### 4.5 CopyPairing Hyperparameter Comparison

As with Copyout we also wanted to compare the different hyperparameter settings for CopyPairing. In particular phase 2 of CopyPairing provides many different options for configuration. Therefore we selected a different ratio of epochs with SamplePairing and Copyout. The results are presented in Figure 5 and show that a ratio of one epoch with SamplePairing and one epoch with Copyout is best (see CopyPairing 1:1).

As a second experiment we investigated if the fine-tuning phase is needed. In this third phase (called fine-tuning) the SamplePairing is normally turned off and just Copyout is used for the last epochs of the training. In this experiment we just skipped this fine-tuning phase and extended the second phase until the end of training. The result is also shown in Figure 5 and indicates that the fine-tuning phase should be retained as it improves the performance.

## 5 Conclusion

In this work the application of *Copyout* and *CopyPairing* is described. Considering the boxplots, we conclude that they are a statistically significant improvement of the state-of-the-art image augmentation techniques *Cutout* and *SamplePairing* (see Section 4.3). But what are the reasons for this improvement?

We conducted several experiments to compare the average magnitude of feature activations as done by Devries & Taylor (2017). But this gave us ambiguous results. That is the reason why we give intuitive explanations. Cutout and Copyout both randomly remove visual features of the training images and forces the network to focus on a wider number of different features to distinguish between different images. This helps to generalize better on unknown pictures. Cutout and Copyout also help to handle occlusion. But what is the relevant distinction between Copyout and Cutout? Why does Copyout perform better?

Cutout performs the occlusion with just a gray square which can be considered wasted space. Copyout fills this space with more visual information which can be observed in Figure 1. One wing of the airplane on the right image is occluded by a frog. This way Copyout not only removes random visual features but also adds new random ones. These random features better correspond to the natural effect of occlusion. In real images for example a truck is more likely not occluded by a grey box but by a car. This helps with further generalization. The small squares of the Copyout technique with additional visual features moreover forces the CNN to differ between important and less important

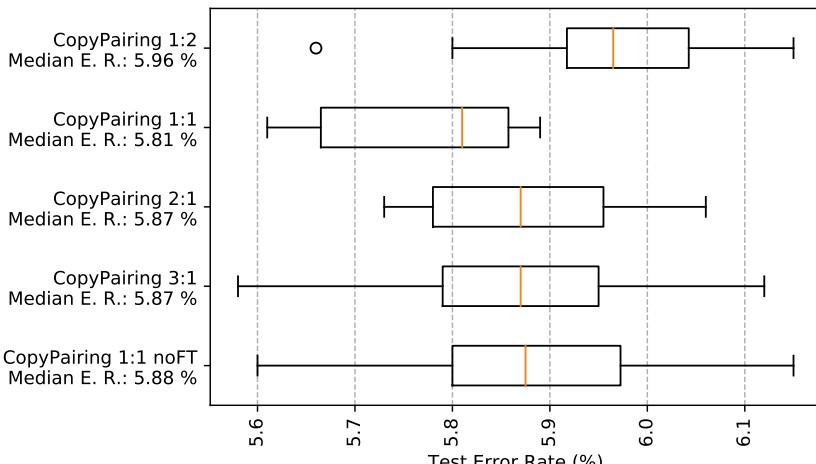

Figure 5: Comparison of different CopyPairing experiments. The ratios of SamplePairing and Copyout epochs have been varied. For example 3:1 means 3 epochs of SamplePairing followed by 1 epoch of Copyout in phase 2 of the SamplePairing technique. CopyPairing 1:1 noFT shows the results of a 1:1 CopyPairing experiment without phase 3 (fine-tuning). Method name with the selected ratio and median test error rate is given by the caption. Each experiment has been repeated at least 12 times to draw the boxplots.

features. The intuitive explanation is that the network learns to focus more closely to the different details to distinguish between important features and misguiding details.

The best presented augmentation scheme is *CopyPairing*. But why does it improve performance compared to SamplePairing? The reason might be that SamplePairing is wasting augmentation potential when it is deactivated for a few epochs in phase 2 and completely deactivated in phase 3 for fine-tuning (also see Appendix A.2). This gap is filled in the CopyPairing technique with using Copyout in-between.

In summary we found two new augmentation techniques that improve the performance for image processing tasks with CNNs. This is especially useful in domains where training data is limited or expensive to obtain as in biomedical applications.

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

## A  APPENDIX

### A.1  KERAS CODE

Listing 1 shows the CNN model definition we used in our experiments. Listing 2 shows the code and parameters for our basic data augmentation. Both is implemented with the Keras neural networks API (Chollet et al., 2015).

Listing 1: Keras CNN Model definition for our Experiments

```
1  input_tensor = Input(shape=(32, 32, 3))
2  x = layers.BatchNormalization()(input_tensor)
3  x = layers.Conv2D(64, (3, 3), padding = 'same')(x)
4  x = layers.Activation('relu')(x)
5  x = layers.BatchNormalization()(x)
6  x = layers.Conv2D(96, (3, 3), padding = 'same')(x)
7  x = layers.Activation('relu')(x)
8  x = layers.MaxPooling2D((2, 2))(x)
9  x = layers.BatchNormalization()(x)
10
11 x = layers.Conv2D(96, (3, 3), padding = 'same')(x)
12 x = layers.Activation('relu')(x)
13 x = layers.BatchNormalization()(x)
14 x = layers.Conv2D(128, (3, 3), padding = 'same')(x)
15 x = layers.Activation('relu')(x)
16 x = layers.MaxPooling2D((2, 2))(x)
17 x = layers.BatchNormalization()(x)
18
19 x = layers.Conv2D(128, (3, 3), padding = 'same')(x)
20 x = layers.Activation('relu')(x)
21 x = layers.BatchNormalization()(x)
22 x = layers.Conv2D(192, (3, 3), padding = 'same')(x)
23 x = layers.Activation('relu')(x)
24 x = layers.MaxPooling2D((2, 2))(x)
25 x = layers.BatchNormalization()(x)
26
27 x = layers.Flatten()(x)
28 x = layers.Dropout(0.4)(x)
29 x = layers.Dense(512, activation = 'relu')(x)
30 x = layers.Dropout(0.3)(x)
31 output_tensor = layers.Dense(classes, activation = 'softmax')(x)
32
33 model = Model(input_tensor, output_tensor)
34
```

```
35  model.compile(loss = 'categorical_crossentropy',
36          optimizer = Adam(),
37          metrics = ['acc'])
```

Listing 2: Keras basic Data Augmentation for our Experiments

```
1  train_image_data_generator = ImageDataGenerator(
2          rescale = 1./255,
3          rotation_range = 15,
4          horizontal_flip = True,
5          zoom_range = 0.2,
6          preprocessing_function = preprocessing_function)
```

### A.2 SAMPLEPAIRING OVERVIEW

Since the CopyPairing augmentation is building on SamplePairing (Inoue, 2018) we give a short overview of SamplePairing. It takes the image it wants to augment, a second random image from the training set and then averages each pixel of the two images. The augmentation is done in these three steps:

1. SamplePairing is completely disabled for the first epochs.
2. SamplePairing is enabled for a few epochs and then disabled again. This is alternating for the majority of the epochs.
3. At the end of training SamplePairing is completely disabled again. This is called fine-tuning.

The SamplePairing paper describes several experiments. For the CIFAR-10 dataset the following configuration is applied: In phase 1 SamplePairing is disabled for the first 100 epochs. Phase 2 takes 700 epochs with SamplePairing enabled for 8 epochs and turned off for 2 epochs alternating. Phase 3 (fine-tuning) lasts for the remaining 200 epochs.

