# OpenReview forum: "Improved Image Augmentation for Convolutional Neural Networks by Copyout and CopyPairing"
_ICLR.cc/2020/Conference — Reject_

### Official Review · AnonReviewer3 · 2019-10-14
**Official Blind Review #3**

**Rating:** 1

**Review:**

I think this paper is not enough to accept in ICLR because
- Lack of novelty.
  - CutMix [1] is very similar to Copyout.
  - To verify the novelty, a more sophisticated description and experimental supports should be required.
- Insufficient experiments for supporting the effectiveness of the proposed method.
  - 6-layer convolutional networks are used, but other architectures, e.g., ResNet, should be demonstrated.
  - Other datasets, e.g., CIFAR100 and ImageNet, should be demonstrated.
  - Various settings, e.g., the number of training samples is limited, should be demonstrated.
  - Need comparison with other augmentation methods, e.g., Mixup, CutMix, AutoAugment.
- Overall, the paper is awkwardly written.

[1] Sangdoo Yun et al. "Cutmix: Regularization strategy to train strong classifiers with localizable features." ICCV 2019.


**Experience Assessment:**

I have published one or two papers in this area.

**Review Assessment: Checking Correctness Of Derivations And Theory:**

I assessed the sensibility of the derivations and theory.

**Review Assessment: Checking Correctness Of Experiments:**

I assessed the sensibility of the experiments.

**Review Assessment: Thoroughness In Paper Reading:**

I read the paper at least twice and used my best judgement in assessing the paper.

---

### Official Review · AnonReviewer2 · 2019-10-23
**Official Blind Review #2**

**Rating:** 1

**Review:**

[Summary]
This paper proposes two data augmentation methods that combine cutout [1] and sample paring [2] for training CNNs, Copyout and Copyparing. The authors evaluate their methods on the CIFAR-10 dataset.

[Pros]
- Data augmentation is an important regularization method for training diverse NN models

[Cons]
- The main issue is novelty. What are the differences of the proposed methods from CutMix [3] and RICAP [4]?
- Only CIFAR-10 was used for evaluation. The results on ImageNet-1k are essential.
- In recent papers, data augmentation methods for training CNN backbones should be validated on various architectures and downstream tasks such as object detection and semantic segmentation.
- The method description is not specific.

[1] Devries and Taylor.  Improved regularization of convolutional neural networks with cutout, ArXiv 2017.
[2] Inoue, Data augmentation by pairing samples for images classification, ArXiv 2018.
[3] Yun et al. CutMix: Regularization Strategy to Train Strong Classifiers with Localizable Features, ArXiv 2019.
[4] Takahasi et al. Data Augmentation using Random Image Cropping and Patching for Deep CNNs, ACML 2018.



**Experience Assessment:**

I have read many papers in this area.

**Review Assessment: Checking Correctness Of Derivations And Theory:**

I carefully checked the derivations and theory.

**Review Assessment: Checking Correctness Of Experiments:**

I carefully checked the experiments.

**Review Assessment: Thoroughness In Paper Reading:**

I read the paper thoroughly.

---

### Official Review · AnonReviewer1 · 2019-10-29
**Official Blind Review #1**

**Rating:** 1

**Review:**

The paper presents an extensions to the CutOut and SamplePairing techniques for image augmentation, CopyOut and CopyPairing. CutOut itself consists of randomly masking out a rectangular region of an image. In CopyOut one chooses a source and target images, and a rectangular region from the source image is copied into target image.


Though the extensions seem to provide an improvement in performance, we feel there are a few improvements that prevent the paper from being accepted:
 - more thorough experimental verification with various CNN architectures and larger variety of datasets datasets is needed (the original paper conducted experiments on CIFAR-10, CIFAR-100, SVHN and STL-10).
- better exposition needed - it would have been helpful to include more examples of the original methods CutOut and Sample pairing. Sample pairing though mentioned is not described. As CopyPairing is largely based on SamplePairing, it would have been helpful to include a diagram or thorough description of it.

**Experience Assessment:**

I have read many papers in this area.

**Review Assessment: Checking Correctness Of Derivations And Theory:**

N/A

**Review Assessment: Checking Correctness Of Experiments:**

I assessed the sensibility of the experiments.

**Review Assessment: Thoroughness In Paper Reading:**

I read the paper at least twice and used my best judgement in assessing the paper.

---

### Decision · Program_Chairs · 2019-12-19

**Decision:**

Reject

**Comment:**

The reviewers have issues with novelty and quality of exposition. I recommend rejection.